# Rodents in Agriculture: A Broad Perspective

**Gary Witmer**

USDA National Wildlife Research Center, Fort Collins, CO 80521, USA; garywitmer51@gmail.com

**Abstract:** There are thousands of rodent species in the world. While they provide a number of ecosystem functions, unfortunately, some species cause significant damage to agriculture. Rodent damage occurs to crops in the field, but also to stored foods, livestock feed, and structures. There are many methods available to reduce rodent populations and/or damage, including both lethal and non-lethal methods. There are advantages and disadvantages to most methods, and many are regulated by federal, state, and local ordinances. Public acceptance of the various methods also varies greatly. Examples and details of these topics are presented in this review.

**Keywords:** agriculture; control; damage; management; rodents; traps; toxicants

## 1. Rodents Background

The largest taxonomic group of mammals is rodents, with over 2200 species known around the world [1]. More recently, it was stated that over 2500 species exist [2]. Many species exist on all continents, with the exception of Antarctica. Rodents have adapted to all ecosystems of the world, including tundra, alpine, temperate forests, grasslands, arid regions, and aquatic systems. They provide many ecosystem functions, including soil aeration and mixing, seed and spore dispersal, vegetation succession, and being an important food source for predatory animals. Some species of rodents are even consumed by people in some parts of the world. Most rodent species are small, secretive, nocturnal, adaptable, and have a keen sense of touch, taste, and smell. Most species have incisors that grow continuously throughout their lifespan, requiring constant gnawing to keep them sharp and at an appropriate length. Many species are active year-round, while some hibernate during the cold winter months, and some estivate during hot and dry summers. Rodent species vary in terms of their reproductive potential, but most are highly prolific, bearing a litter of several young every year with some even producing several litters per year. Some rodent species live relatively long lives (k-selected species), but most live short lives (r-selected species). Some species maintain stable populations, but some show peaks and valleys in population sizes [3]. This occurs with vole (*Microtus*) populations in the northern hemisphere, which peak every 3–5 years and crash thereafter. This is often related to weather and vegetation conditions, but also predatory pressures [4]. While most rodent species are relatively small and non-descript, they can vary widely in size from mice that weigh 10 g to capybara (*Hydrochaerus*) which can weigh 70 kg. A much broader background on rodents, including their evolution, morphology, diversity, social organization, behaviour, and ecology, has been reviewed [5].

## 2. Rodent Damage

Most rodent species do not cause significant damage to human resources and property, but about a handful of genera on each continent do [6,7]. These include:

- North America: Cynomys, Geomys, Marmota, Microtus, Spermophilus, Thomomys;
- Central and South America: Ctenomys, Holochilus, Octodon, Oryzomys, Sigmodon, Zygodontomys;
- Europe: Apodemus, Arvicola, Clethrionomys, Critetus, Microtus;
- Africa: Arvicanthis, Gerbilliscus, Mastomys, Meriones, Rhabdomys, Thryonomys, Xerus;

- Middle East: Hystrix, Meriones, Microtus, Nesokia, Psammomys, Spalax;
- South Asia: Arvicola, Bandicota, Spermophilus, Cricetus, Meriones, Microtus, Nesokia, Rattus, Tatera;
- Southeast Asia/Pacific Islands: Bandicota, Callosciurus, Mus, Rattus;
- Australia: Rattus, Mus, Melomys.

It is important to note that other mammals can cause crop damage, including birds, rabbits, and ungulates [8]. Almost all crops grown around the world can be damaged by rodents, including cereal grains, vegetables, cotton, alfalfa, sugar cane, potatoes, tree fruits, and many others [7]. Damage can occur during any part of the plant life cycle. It can also occur during any part of the year, including under snow cover. Rodents will also shift their feeding preferences during the plant life cycle, focusing on planted seeds, then germinating plants, and finally mature plants and their seeds/fruits. Most rodents prefer succulent vegetation, but many will shift to senescent vegetation or woody vegetation, including bark and roots as the year progresses. Some subterranean rodent species (e.g., pocket gophers, *Thymomys*) feed almost entirely on underground plant material [9]. Damage levels in crops can vary from insignificant (a few percent) to very significant (>30%) to almost complete crop loss; see examples from around the world have been presented [6]. Damage levels from commensal rodents (*Rattus*, *Mus*) are generally less that those from native rodent species, but still can amount to 1–15% and can be even higher on some islands [10]. As an example of the severe levels of rodent damage that can occur, it was reported that in Asia alone, the amount of grain eaten by rodents would provide enough food to feed 200 million people for a year [11]. Also, damage levels can vary greatly from year to year. Crops in storage can also be damaged by rodents along with livestock feed. This occurs from direct consumption, but also by contamination from urine and feces. Other types of damage from rodents comes from their digging/burrowing which can undermine foundations and damage farm equipment and/or livestock passing by. Rodents can also damage wiring and insulation in buildings. Finally, rodents can transmit various diseases to people and livestock, including plague, leptospirosis, chagas disease, giardiasis, and hantavirus, among many others [12,13].

## 3. Rodent Damage Control

Humans have developed and used many methods to reduce rodent damage to crops [6,14]. Additionally, methods are being improved and new methods developed all the time. Methods include both lethal and non-lethal methods. The methods that are most commonly used vary around the world. In many locations, the methods that can be used are regulated by various federal, state, and/or local government agencies; however, people often use whatever is convenient, inexpensive, and/or familiar to them.

Non-lethal methods include **habitat management**, both in the crop fields, but also in surrounding vegetation. Those surrounding areas often provide cover and food for rodents year around, but also act as refugia for rodents after crops are harvested where they can survive until the next crop cycle begins. After a crop harvest, crop land can be made less habitable to rodents by burning or otherwise removing plant stubble. Unfortunately, that can also result in increased soil erosion and/or water loss. The surrounding "refugia" areas can also be mowed or burned so that they are less supportive of rodents. Less palatable plants can be used on lands surrounding crop fields, such as endophytic grasses which contain a fungus unpalatable to herbivores [15]. Arid and semi-arid lands generally do not support a lot of rodents, but when we irrigate those lands for crop production, we greatly increase their rodent carrying capacity. Another way to modify habitats is to install raptor perches which can result in greater predation on rodents and other small mammals such as rabbits [16].

Other non-lethal approaches include the use of **barriers** to limit access by rodents [17]. This often involves use of wire-mesh fencing around the crop field. Unfortunately, this is expensive and also requires considerable maintenance. Care must also be taken to not damage the fences with farming equipment such as tractors. In some countries a low

electric wire has been placed around crop fields. This has the same problems as wire-mesh fencing, plus the risk of starting fires.

Rodent **frightening devices** have also been used, both inside buildings and in fields [18]. In buildings, these devices include ultrasonic devices, other noises, and flashing lights. Some of the field devices are put into the ground and produce a vibration meant to discourage burrowing rodents. In most studies, these devices have been found to be rather ineffective; if nothing else, rodents usually adapt/acclimate to the devices over time [19].

Some **repellents** have proven effective against rodents, and a few are registered for such uses [18]. One of these on the market is capsaicin, the active ingredient in hot peppers, but this is rather expensive and requires a fairly high concentration ($\geq 2\%$) to be effective. Sulfurous odors from predator feces or urine can also be effective. Some plant secondary metabolites have also shown promise as rodent repellents [20]. Unfortunately, these are not practical for use over large areas such as crop fields and also may cause issues such as flavor problems with human foods. Of course, rodents can adapt/acclimate to them, especially if hungry enough.

A relatively new rodent control method is **fertility control** [21]. This involves putting out a palatable bait or liquid that contains a drug/chemical that reduces or prevents the rodent from being able to breed successfully. Some of these chemicals include diazacon, nicarbazin, vinylcyclohexene, and triptolide [22,23]. Additionally, genetically modified rodent strains have been developed that can be placed on islands and can breed and produce sterile offspring [24]. One fertility control product, ContraPest, was recently registered by the US EPA for use in black rat and Norway rat control [22]. This is a liquid formulation containing vinylcyclohexene and triptolide [22,25]. These approaches can greatly reduce a rodent population fairly quickly and, hence, crop damage. In theory, the rodent population would die out over time, but this would probably not happen due to immigration from surrounding areas. On the other hand, eradication might be achieved for invasive rodents on islands where there is not a ready source of immigrants [24].

**Lethal methods** include traps, shooting, flooding, and the use of toxicants (rodenticides) [14]. Again, it needs to be mentioned that the use of most of these methods are regulated by various regulations and laws. Lethal methods also come with considerable controversy with the public.

Many **traps and trapping methods** have been developed and used for mammals [26], including leghold traps used for furbearing predators and some larger rodents. Additionally, many kill traps have been developed and are available for rodent control [14]. Although there are various designs and sizes, most kill by body constriction. The use of traps is labor intensive because they have to be placed, "set", and checked periodically. Additionally, once sprung (whether they have caught a rodent or not), they must be re-set. Various baits are used to attract rodents to traps and it is important to identify and use a very attractive bait for a specific location and setting [27]. More recently, some self-resetting multiple kill traps have been developed which may make kill traps more efficient to use [28]. It should be mentioned that non-kill traps ("box" traps) are also available but are rarely used outdoors. Some people prefer to use them indoors, but the captured rodents still need to be dealt with. Multiple-capture live traps have been available for house mice and have been developed and tested for some larger rodents such as nutria (*Myocastor*) [29]. An advantage of live traps is that non-target animals can often be released unharmed.

The use of **oral rodenticides** is very popular around the world, although their use is not without much controversy and disagreement [30,31]. Rodenticides meant for consumption (i.e., oral rodenticides) comprise two main categories: acute rodenticides are very potent and generally kill the rodent fairly quickly after a single feeding of a small amount. These include alpha-chloralose, bromethalin, cholecalciferal, red squill, sodium fluoroacetate, strychnine, and zinc phosphide [14]. On the other hand, anti-coagulant rodenticides require more to be eaten and the rodent dies relatively slowly from internal and external bleeding. First generation anti-coagulants include warfarin, chlorophacinone, and diphacinone; however, "second-generation" anti-coagulants have been developed which

act more quickly and require much less to be consumed. These include bromadialone, brodifacoum, difenacoum, flocoumafen, and difethialone. There is a lot of concern among the public about the inhumaneness of anti-coagulants. Rodenticides are not rodent-specific, so non-target animals can be killed directly by feeding on the toxicant (primary poisoning), or by feeding on poisoned rodents (secondary poisoning). Reducing non-target animal hazards can be done by taking care in where the rodenticides are placed, but also by using bait boxes so that larger birds and mammals cannot access the toxic bait. It should be noted that while acute toxicants pose a primary hazard to any animal that consumes it, they generally do not pose a secondary hazard to animals that consume dead rodents as the rodent has metabolized the toxin or it has otherwise been dissipated.

**Fumigants** are another type of rodent toxicant that can be used in some situations [14]. These work by the rodent breathing in toxic fumes. Fumigants (e.g., methyl bromide) are used in buildings to rid them of rodents and insects. Care must be taken because the gases are very toxic to humans and domestic animals. Fumigants can also be used to kill burrowing rodents. The fumigant is placed in the burrow which is then sealed with soil and/or paper so that the gas can stay in the burrow long enough to kill the resident rodents. Burrow fumigants include aluminum phosphide, magnesium phosphide, acrolein, carbon/sodium nitrate, and propane. Like oral toxicants, burrow fumigants are not species-specific and will kill any animals in the burrow. All fumigants are regulated by various governmental agencies.

Many landowners do not respond to rodent presence until damage is very evident. Rodent populations can still be reduced at this point, but the damage and losses cannot be retrieved. Hence, it is a good idea to monitor rodent populations periodically so that control measures can be implemented before damage becomes significant. **Monitoring rodent populations** can be done in various ways, but is mainly done with the use of transects or grids. Along transects, the person stops periodically, e.g., every 10 m, and looks for fresh rodent sign (tracks, droppings, digging, vegetation gnawing). Grids are done similarly at each grid stop location. There is no definitive percent of rodent "positive" stations that should cause a person to initiate rodent control, but some researchers and managers recommend the control be initiated at greater than 10% rodent-active sites. More intense and costly monitoring requires the use of traps or non-toxic chew blocks being placed at each monitoring station. Field cameras are now using for monitoring many types of wildlife, and while they may be used for larger rodent monitoring [29], the small body size of many rodent species may limit effectiveness. Greater detail and other methods of population monitoring can be found in [32].

## 4. Rodent IPM

Integrated pest management (IPM) has long been applied in plant and invertebrate pest management [33]. It has not been used as much in vertebrate pest management. Most often, the convenient, efficient, and effective method is used for rodent damage control. For larger vertebrates such as predators and ungulates, this has involved shooting and traps. For rodents, it has involved the use of rodenticides or traps. This continues despite the fact that rodents can adapt/adjust in various ways when a single control method is used. They can become unfazed by frightening devices and repellents over time. In the case of toxicants, they may become physiologically or genetically unaffected by these methods. This occurred in the case of the heavy use of first-generation anti-coagulants whereby rodents became resistant to them. This led to the development and use of second-generation anticoagulants [29].

Rodent IPM requires careful consideration and integration of three areas: (1) Rodent population management, involving consideration of the biology, population dynamics, and ecology of the pest rodent; (2) habitat management, involving consideration of the physical and biotic environment; and (3) people management, involving consideration of land uses and management, and human activities [6]. Fortunately, decision support systems have been developed for rodents that can help with the incorporation of all these factors [6].

Rodent population and damage management has less often involved the use of preventive methods and non-lethal methods, nor a combination of methods, despite the fact that a combination of methods may prove to be more effective and acceptable to the public [34]. Sanitation management in the general agriculture area can also reduce crop losses. This involves not having protective cover areas around, such as brush piles and rock piles, and not having human food waste, livestock feed, and pet food available to rodents. Combining some rodent barriers with increased sanitation could further reduce damage. For many other suggestions, including "ecologically-based" rodent management, see the review [11].

## 5. Conclusions

Despite the many tools and methods available to control rodent populations and damage, we still see substantial damage by rodents around the world. It is safe to assume that this will continue especially when we provide abundant and nutritious food for them along with protective cover. In fact, the significance of these crop losses will increase in severity as the human population continues to grow and climate change impacts (e.g., drought) increase. Research on rodent control methods must continue, if not increase, to improve existing methods and develop new methods. These improvements could include, but are not limited to, detection methods, new rodenticides, effective repellents, barrier effectiveness and durability, biological control, fertility control, and habitat manipulation. Additionally, safer methods are needed to deliver rodenticides such that the risk of exposure to people, pets, livestock, and wildlife can be further reduced. Efforts to incorporate public education are also needed so that the public will better understand rodent populations and the damage they cause, but also that they will be more accepting of control measures. Hopefully, this will reduce the restrictions and bans on rodent control measures that seem to be coming more often.

**Funding:** This research received no external funding.

**Data Availability Statement:** Not applicable.

**Acknowledgments:** The author acknowledges the many colleagues he worked with during his career.

**Conflicts of Interest:** The author declares no conflict of interest.

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
