# Peer review of "Rodents in Agriculture: A Broad Perspective"

_agronomy, doi:10.3390/agronomy12061458_

Round 1

Reviewer 1 Report

Gary Witmer's manuscript is a useful summary of the broad issues of rodents and agriculture. The work fits exactly into the Special Issue ‘Rodents in Crop Production Agricultural Systems’ of the ’Agronomy’ journal. I appreciate the English of the manuscript is easy to understand even for me as a non-native speaker. The text familiarizes a reader quickly and clearly with this complex and changing topic. At the same time, thanks to the extensive bibliography, it can serve as a guide to references to partial aspects of the topic. Literary sources are adequate in number and quality.

Nevertheless, I have a few suggestions that I believe could improve the resulting article. They are listed chronologically, resp. in the order in which they appear in the text.

Chapter 1 (Rodents background)

- In the first sentence of the first paragraph, the author states that there are over 2,200 species of rodents in the world (Nowak, 1999). Recent work, however, reports an even higher number; Pardiñas, U. F. J., et al. from 2016, resp. 2017 lists 2475 species (Pardiñas, U. F. J., et al., 2017: Handbook of the Mammals of the World, Volume 7. Rodents II); I recommend referring here to Burgin et al., 2018, where even over 2500 species are already mentioned (Burgin et al. 2018; Journal of Mammalogy, 99 (1): 1-14, 2018).

- I would recommend replacing the term ‘non-descript’ in the penultimate sentence with another.

Chapter 2 (Rodent Damage)

- the list of areas would be good to sort alphabetically, i.e., from Africa. It would also be good to sort the lists of rodent genera in each area consistently in alphabetical order (applies to at least South Asia and Australia)

- with respect to the current taxonomy, the genus 'Pitymys' should disappear. It is usually already considered part of the genus ‘Microtus’. In the list of African species, the genus 'Tatera' should be replaced by the genus 'Gerbilliscus' (both according to Pardiñas, U. F. J., et al.)

- I assume a typo in the genus 'Cricetus' (not 'Critetus')

- at least in Central Europe, rodent damage is also significant on fruit trees, so I would explicitly mention them in the list of crops

- I assume a typo in the sentence ‘Damage levels from commensal rodents (Rattus, Mus) is generally less than that from native rodent species, but still can amount to 1-15% and can be even higher on some islands [9].’

Chapter 3 (Rodent Damage Control)

- My main comment is on this chapter layout. I would prefer greater clarity. My favourite solution would be subheadings. Other options include a first paragraph with a brief summary/list of the chapter topics, or a keyword highlighting in each paragraph.

- Paragraph 5: The situation with repellents is locally different, perhaps even more problematic in Europe (e.g. Stefanini, M., Charon, M., & Marchand, P. A. (2020)) Rodent repellents at a European Union Plant Protection Product level, Journal of Plant Protection Research, 1-6.) I recommend extending this to a non-US context.

- I would appreciate extended paragraph on rodenticides. It could be structured as follows: What are rodenticides? What groups are they divided into (acute and anticoagulant)? Which substances were / are used and to which group do they belong? What are the advantages and disadvantages of each group (e.g. in terms of speed and duration of action, ‘welfare’, applicability to different groups of rodents (e.g. social and solitary), risks to non-target organisms, persistence of residues, etc.

- Missing word: ‘It should be noted that while acute toxicants pose a primary hazard to any animal that consumes it, they generally do not pose a secondary hazard to animals that consume dead rodents as the rodent has metabolized the toxin or it has otherwise been dissipated.’

Chapter 4 (Rodent IPM)

-No comment

Chapter 5 (Some Concluding Thoughts)

- ‘…new methods…’

The extent of changes fit the criteria of a minor revision.

Author Response

Reviewer 1 liked the existing manuscript and only mage a few suggestions for improvements. I have incorporated those in the new version being submitted. These include updating the number of rodent species and adding that citation. I also corrected some species names in the listing. A few typos were corrected. I also put control method names in bold so that stand outbetter.

Reviewer 2 Report

This paper reviews the importance of rodents to agriculture. The paper looks like an introductory text to some special issue devoted to rodents. If this is not so, then its publishing makes no sense to me. In general, it is nicely written and organised. I do not have any problem with it. I did not find any serious deficiencies, errors, or mistakes. The following comments are just offered for further consideration.

1)      The paper deals just with the negative effects of rodents to crop. However, I believe rodents have positive effects as well, e.g., on soil fertility. This is not reflected in the title.

2)      When introducing rodenticides, it starts with the acute ones, making the reader expect that the chronic rodenticides will be treated afterward. Only those who know that anti-coagulants are among chronic ones will understand.

3)      Primary hazards were reduced considerably when grain baits were replaced by pelletised baits, particularly those with no bonding agent decaying within a few days after treatment.

4)      When addressing monitoring, the authors might focus not only on methods but also on quantitive processing of data towards the models forecasting population densities in the next year as a regular part of IPM.

5)      The concluding thoughts are not linked to the rodents or damage in the introductory sections. Instead, the authors consider progress in rodent control in general. They are very weekly critical to the present situation in rodent control which becames for farmers worse and worse due to increasing numbers of restrictions and limitations. In Europe, rodent pest control has become very difficult these days.      

Author Response

Reviewer 2 liked the original manuscript and offered a few suggestions, but none of these seemed necessary nor would they improve the existing wording.